## [Decision Letter · Decision Letter 0]

26 Aug 2025

HIV-1 Vpr is a novel autophagy target and its susceptibility to autophagy reduces virus transmission

PLOS Pathogens

Dear Dr. Serra-Moreno,

Thank you for submitting your manuscript to PLOS Pathogens. After careful consideration, we feel that it has merit but does not fully meet PLOS Pathogens's publication criteria as it currently stands. Therefore, we invite you to submit a revised version of the manuscript that addresses the points raised during the review process.

Please submit your revised manuscript within 60 days Oct 25 2025 11:59PM. If you will need more time than this to complete your revisions, please reply to this message or contact the journal office at plospathogens@plos.org. Please include the following items when submitting your revised manuscript:

We look forward to receiving your revised manuscript.

Kind regards,

Edward M Campbell, PhD

Academic Editor

PLOS Pathogens

Susan Ross

Section Editor

PLOS Pathogens

Editor-in-Chief

PLOS Pathogens

orcid.org/0000-0003-2946-9497

Editor-in-Chief

PLOS Pathogens

orcid.org/0000-0002-7699-2064

**Journal Requirements:**

1) Please provide an Author Summary. This should appear in your manuscript between the Abstract (if applicable) and the Introduction, and should be 150-200 words long. The aim should be to make your findings accessible to a wide audience that includes both scientists and non-scientists. Sample summaries can be found on our website under Submission Guidelines:

https://journals.plos.org/plospathogens/s/submission-guidelines#loc-parts-of-a-submission

2) We have noticed that you have uploaded Supporting Information files, but you have not included a list of legends. Please add a full list of legends for your Supporting Information files after the references list.

3) Some material included in your submission may be copyrighted. According to PLOSu2019s copyright policy, authors who use figures or other material (e.g., graphics, clipart, maps) from another author or copyright holder must demonstrate or obtain permission to publish this material under the Creative Commons Attribution 4.0 International (CC BY 4.0) License used by PLOS journals. Please closely review the details of PLOSu2019s copyright requirements here: PLOS Licenses and Copyright. If you need to request permissions from a copyright holder, you may use PLOS's Copyright Content Permission form.

Potential Copyright Issues:

i) Figure S5. Please confirm whether you drew the images / clip-art within the figure panels by hand. If you did not draw the images, please provide (a) a link to the source of the images or icons and their license / terms of use; or (b) written permission from the copyright holder to publish the images or icons under our CC BY 4.0 license. Alternatively, you may replace the images with open source alternatives. See these open source resources you may use to replace images / clip-art:

ii) We note that Figures 10 and S4 are created through BioRender. Please confirm that you hold a Premium account and provide a pdf copy of the CC BY 4.0 Licence as provided by BioRender. For instructions on how to generate a CC BY 4.0 license for your figure, please see the guidelines here: https://help.biorender.com/hc/en-gb/articles/21282341238045-Publishing-in-open-access-resources.

If you are using the free assets from BioRender, we are unable to publish these images as they are licenced under a stricter licence than CC BY 4.0. In this case we ask you to remove the BioRender images and replace them with open source alternatives.

See these open source resources you may use to replace images / clip-art:

- https://bioart.niaid.nih.gov/

- https://bioicons.com/

- https://healthicons.org/

- https://scidraw.io/

- https://reactome.org/icon-lib

- https://www.phylopic.org/images

- https://journals.plos.org/plosbiology/article?id=10.1371/journal.pbio.3002395

4) We note that your Data Availability Statement is currently as follows: "All the data is found in the manuscript and the supplementary materials". Please confirm at this time whether or not your submission contains all raw data required to replicate the results of your study. Authors must share the “minimal data set” for their submission. PLOS defines the minimal data set to consist of the data required to replicate all study findings reported in the article, as well as related metadata and methods (https://journals.plos.org/plosone/s/data-availability#loc-minimal-data-set-definition).

3) If any authors received a salary from any of your funders, please state which authors and which funders.

6) Please ensure that the funders and grant numbers match between the Financial Disclosure field and the Funding Information tab in your submission form. Note that the funders must be provided in the same order in both places as well.

**Reviewers' Comments:**

Reviewer's Responses to Questions

**Part I - Summary**

Reviewer #1: This study from the Serra-Moreno group is an elegant demonstration that Vpr-derived from lab adapted HIV-1 NL4-3 is an autophagy target itself, whereas transmitted-founder virus derived Vprs are resistant towards autophagy. By a series of convincing experiments, the authors show that NL4-3 Vpr is sensitive towards degradation when autophagy is induced and this is true in Jurkat CD4+ T cells as well as in primary CD4+ T cells. They show localization of Vpr to autophagosomes; by sequence alignment with T/F Vprs and comprehensive mutagenesis they identify residues in Vpr that confer resistance to autophagy sensitivity of Vpr. Of note, lab-adapted NL4-3 Vpr interacts with autophagy receptors in contrast to T/F virus-derived Vpr. Ubiquitination of Vpr does not seem to be a determinant of Vpr-depletion via autophagy. However, Vpr-sensitivity towards autophagy might be a determinant that lowers efficient HIV-1 transmission in their models.

Altogether, the authors provide a set of convincing results that largely support the conclusions. The strength of the manuscript is the consequent quantification of viral protein expression, i.e. Vpr levels in the presence/absence of autophagy inhibitors in CD4+ T cells (Jurkats) and primary CD4+ T cells. Furthermore, the mutagenesis approach and reverse genetics to delineate the exact residues conferring Vpr-resistance to autophagy are elegant and convincing.

In my opinion less clear remains the phenotypic characterization in terms of biological relevance/HIV-1 transmission (figure 10, see specific comments below). Furthermore, I would like to see some more controls in order to corroborate certain aspects of the manuscript.

Story-telling wise, while I really enjoyed reading the results section, the introduction should be shortened, since it virtually is repetitive to the abstract and pre-introduces all the results. The discussion is again largely redundant to the results and mainly re-iterates them, instead of discussing the potential broader relevance of the findings. In this regard, I do not see any realistic therapeutic application of autophagy inhibition in the field of HIV antivirals. We are in the lenacapavir era, so do you really want to argue somebody would develop an autophagy exploiting approach to target HIV-1? Please tune down these statements.

Specific comments:

1. Figure 2A/D; I do not agree that rapamycin and torrin2 cause a dose dependent reduction in Vif levels

2. Figure 4B/C: Can you perform complementary microscopy to potentially demonstrate that Nef displaces Vpr from autophagosomes? Are T/F Vprs not located in autophagosomes? Upon introduction of autophagy-resistance conferring mutations in Vpr, does it not localize to autophagosomes anymore?

3. Figure 7G: The “further increase” in Vpr level upon NDP52 KO is not convincing; perform signal quantification via multiple WBlots as done before and I would suggest to include an NDP52 single KO to corroborate this

4. Figure 10: Unfortunately, you missed to include a delVpr variant in these experiments throughout to narrow down the relative importance of the autophagy-resistance versus other Vpr functions, respectively a loss of Vpr-expression;

Furthermore, in my opinion, there is a logical gap. As far as I can see the employed constructs are Nef-intact and the authors argue themselves that NL4-3 Nef counteracts autophagy in a way that Vpr-sensitivity towards autophagy is not a determinant, especially in Jurkat CD4+ T cells (Figure 1A). Therefore, the authors should include a Nef-defective version of their constructs in these assays. On top, since they elegantly identified Vpr-mutations conferring resistance to autophagy it would have been straight forward to test NL4-3 Vpr vs NL4-3 Vpr I37P H45Y (that is resistant to autophagy). This would allow to more directly conclude on autophagy as an underlying cause for potential differences as compared to a benchmark of NL4-3 Vpr vs the T/F CH077.

5. line 495: careful, as far as I am aware of none of these observations were clearly confirmed in primary T cells or even Jurkats. Other non-yet identified Vpr-functions might facilitate or enhance transmission in their systems. It would have been very nice to include primary CD4+ T cells in their models in figure 10; however I appreciate that this goes beyond what one could expect at this stage. I strongly encourage the authors to discuss the Jolly paper in this regard (PMID: 35417711).

6. line 581: again, if Nef of lab adapted NL4-3 inhibits autophagy, especially in cell lines, this argumentation does not make sense

7. the authors do not really discuss or address potential underlying reasons for discrepant results compared to an earlier study that demonstrates modulation of autophagy by Vpr (PMID: 31628772). Different cell systems or Vpr alleles can not serve as an explanation (Reference 45 uses also NL4-3 Vpr and Jurkat cells, argumentation in lines 500 – 503).

8. Apparently, the broader relevance and biological significance of the authors findings is limited, as Vpr-sensitivity towards autophagy seems to be the exception from the rule. However, as most studies employ HIV-1 NL4-3 and NL4-3 Vpr, it is very important to stress this potential confounding factor in studies trying to find host cell factors that are depleted by Vpr. The authors might want to discuss this and raise awareness to such limitations imposed by autophagy sensitivity of NL4-3 Vpr.

Reviewer #2: This manuscript investigates the degradation of HIV-1 Vpr by autophagy and claims that autophagy resistance of Vpr in transmitted/founder (TFV) viruses promotes transmission fitness, in contrast to lab-adapted strains like NL4-3. While the topic is conceptually interesting, the work suffers from serious gaps in experimental rigor, overreaching conclusions, and insufficient mechanistic resolution. The central problem is the incomplete and sometimes misleading portrayal of autophagy dynamics—particularly whether Vpr is simply a passive target of autophagy or also an active modulator of the pathway.

Reviewer #3: In this submission, Chen et al. report HIV Vpr as a novel substrate/target protein of the macroautophagy pathway, meaning that a subpopulation of Vpr in cells is degraded in an autophagosome-dependent pathway involving autophagosome-lysosome fusion (i.e. macroautophagy). The authors provide mostly clear findings that autophagy induction by mTOR inhibitors results in a certain degree of Vpr protein loss, in support of their main conclusion. Significantly, the authors show that knockout of autophagic receptors results in recovery of Vpr protein following mTOR inhibition. Furthermore, the authors show that certain lab-adapted Vpr proteins (like that of NL4-3) are more susceptible to turnover by autophagy relative to Vpr proteins encoded by transmitted/founder viruses. This suggests that there may be selective pressure for Vpr to avoid recognition and destruction by the autophagy machinery. I mostly appreciated how the manuscript was written and how the authors comprehensively cover the previous literature. However, I think the manuscript could be significantly improved with modifications to the text and experimental additions.

**Part II – Major Issues: Key Experiments Required for Acceptance**

Reviewer #1: Experiments I suggested to improve data shown in Fig.4B/C, Fig.7 and Fig.10

Reviewer #2: General Critique and Central Hypothesis

- The authors claim that HIV-1 Vpr is a direct target of autophagy, and that resistance to autophagy promotes viral transmission.

- Most experiments rely on overexpressed Vpr or non-replicating systems, not full-length HIV-1 infection.

- Overexpression artifacts are a major concern, particularly in HEK293T cells, which do not represent natural infection environments.

- The hypothesis that autophagy-sensitive Vpr correlates with reduced transmission is speculative and unsupported by causal data.

- The claim that Vpr sensitivity to autophagy is a hallmark of lab adaptation lacks evolutionary justification.

Experimental Strategy and Key Technical Concerns

- Torin2, a non-specific mTOR inhibitor, upregulates both autophagy and proteasome activity, complicating the attribution of Vpr degradation to autophagy.

- The use of MG132 does not adequately rule out proteasomal involvement; more rigorous assays (e.g., proteasome activity reporters) are lacking.

- Autophagy-deficient models (ATG5 KO and VPS34 inhibitors) were used, but flux analysis was not performed.

- p62 and LC3-II/I measurements are insufficient without flux controls (e.g., Bafilomycin A1).

- There is a lack of experiments in physiologically relevant cell types such as macrophages or primary T cells.

Figure-by-Figure Critique

- Figures 1–3: Vpr degradation is observed in autophagy-competent cells, but flux validation and pathway specificity are not demonstrated.

- Figures 4–5: The link between Vpr degradation and reduced transmission is correlational. No rescue experiments or mechanistic proof are provided.

- Figures 6–7: Residue mapping is performed in overexpression systems, not full virus contexts. Structural modeling is superficial. The role of residues in full HIV-1 replication remains untested.

Conceptual and Logical Gaps

- Equating Vpr's presence in LC3+ compartments with autophagic degradation is an overreach; other routes (e.g., lysophagy or direct proteasomal degradation) are not excluded.

- Claims about lab-adapted vs. primary isolate Vpr are not well-supported; the phylogenetic and sequence analysis is shallow.

- Structural mapping lacks functional testing in full-length virus.

- No autophagy receptors, ubiquitination signals, or degradation adaptors are identified.

- Torin2’s dual activation of proteasome and autophagy was not experimentally separated, undermining conclusions.

Recommendations and Final Evaluation

- Include autophagic flux assays using Bafilomycin A1 or Leupeptin.

- Use live-cell reporters for Vpr degradation and full-length HIV-1 constructs for testing mutant phenotypes.

- Perform infectivity and transmission assays with autophagy-deficient target cells.

- Rescue experiments (e.g., using autophagy-resistant Vpr) are needed to link degradation to viral transmission.

- The hypothesis is interesting but insufficiently supported. Mechanistic clarity and evolutionary context must be strengthened.

- Recommendation: Major revision.

Reviewer #3: 1. Figure 1: LC3 immunoblotting results are confusing. Autophagic flux is generally measured by assessing LC3-II levels, since LC3-II is a hallmark of autophagy initiation, with losses to LC3-II suggestive of clearance in the lysosome following autophagosome-lysosome fusion. Why is LC3-I selectively lost upon rapamycin treatment? Can the authors show that bafilomycin A1 restores LC3-I in rapamycin treated cells? It may be useful to do LC3-GFP cluster analysis in this figure to complement immunoblotting. Also, the results in the primary CD4+ T cells suggest that LC3-II levels are high under basal conditions and rapamycin does not impact LC3-II levels. This is an indication of incomplete autophagic flux (i.e. a defect in autophagy), which doesn’t support the authors claims of “primary CD4+ T cells have more active autophagy at steady state.” Authors should include some instances where mTOR inhibition and bafilomycin are combined and modify how they discuss these results. This is important considering the previous publication reporting that Vpr inhibits autophagic flux (Alfaisal et al, 2019 and others).

The authors should include an experiment in which, in an experiment with increasing concentrations of rapamycin/Torin2, they include a rapamycin + bafilomycin A1 condition. Restoring Vpr with bafilomycin A1 (which inhibits autophagosome-lysosome fusion) would further support the authors’ claims that Vpr is targeted by macroautophagy. Also, the rapamycin + bafilomycin A1 condition would be helpful for addressing LC3 and SQSTM1 phenotypes. Bafilomycin A1 will prevent autophagosome-mediated turnover of LC3-II, and the authors should expect to see LC3-II levels increased compared to rapamycin alone. The combination of conditions would then allow the authors to properly speak about autophagic flux (flux = turnover). Without restoring LC3-II with bafilomycin A1 in mTOR-treated cells, the authors can only speak of LC3 lipidation levels, not autophagic flux. The authors use the term “maturation” to imply that autophagosome-lysosome fusion has occurred, but I would argue that “flux/turnover” is the correct term, and that term can only be used when the extent of turnover is revealed by including bafilomycin A1 or a similar drug. Similarly, an mTOR inhibitor + bafilomycin should be added to immunofluorescence data, because Vpr expression may intensify in that condition and it would reveal the location of Vpr prior to its terminal degradation in lysosomes.

2. Line 216: chloroquine, as an inhibitor of lysosome acidification, is expected to inhibit macroautophagy, since lysosomes are an integral part in macroautophagy. In fact, there is no “autophagy” without lysosomes. Autophagosomes must fuse with lysosomes in order for cargo to be degraded. How can the authors explain why chloroquine does not restore Vpr levels? Use of bafilomycin A1 could also be added, which like chloroquine, should disrupt lysosomal activity and therefore inhibit autophagosome-mediated protein turnover (autophagy). Also, there is no loss of Vpr protein upon Torin2 treatment in Supplemental Figure 1, so the authors are unable to say whether chloroquine or MG132 rescued Vpr protein levels.

3. Figure 10: The authors could have tested the impact of the specific residues in Vpr that govern sensitivity to turnover by autophagy on HIV transmission in their 2D and 3D assays, instead of comparing NL4-3 Vpr to T/F Vpr. As the data currently stands, the authors cannot distinguish between the impact of multiple Vpr functions, which are likely to be different between NL4-3 Vpr and T/F Vpr. Since the authors have engineered full length HIV encoding mutant Vpr, the authors should use the chimeras/mutants of NL4-3 Vpr and CH077 Vpr to specifically address whether residues that dictate susceptibility to autophagic turnover and binding to autophagy receptors are important for HIV transmission. As it stands, the data does not report large effect sizes and is hard to connect with the other figures. Does NL4-3 Vpr transmit better in these 2D and 3D culture systems when the C-terminal residues at “ISTR” from CH077 have been introduced into it? Also, the authors could have included additional Vpr mutants which have been shown to disrupt other, previously recognized, functions of Vpr. This would allow the authors to discern just how important the autophagy-dependent turnover of Vpr is to virus spread, compared to other Vpr functions.

**Part III – Minor Issues: Editorial and Data Presentation Modifications**

Reviewer #1: see my comments related to the introduction and discussion part of the manuscript

Reviewer #2: (No Response)

Reviewer #3: 1. There are redundancies in the Introduction and Discussion, including topics and references. I would remove those from the Discussion and focus more on the implications of your work rather than rehighlighting critical references that were already mentioned in the Introduction.

2. The sentence at line 59 is confusing and should be re-written: “As class III PtdIns3K complex II produces PtdIns3p.” It’s unclear whether the authors are claiming that complex I or complex II (or both) is producing PtdIns3P. Perhaps replace “as” with “like.”

3. Line 103: replace “would” with “may.”

4. Line 106: Begin sentence with, “In this manuscript…”

5. mTOR should be written as “mTOR” and not “MTOR” as per standard convention in the literature.

6. Figure 2: the loss of SQSTM1 in PAR by western in Figure 2A is not apparent in the membranes depicted, making it hard to envision how the quantified relative SQSTM1 levels were quantified in Figure 2C. Also, instead of the term Parental (PAR), I would use the term WT for clarity. You can specify “parental, WT” cells in the text and then write “WT” in the figure.

7. Figure 2D: can the authors perform a similar experiment, but using SAR405? SAR405 was shown to be a specific vps34 inhibitor and any results obtained with it could corroborate findings using VPS34IN.

8. Figure 4B-C: it seems that Nef expression, in the absence of mTOR inhibitor, causes increased levels of Vpr protein. Can the authors comment on this and discuss its significance? I do not find these results particularly convincing, nor are they essential to the message of the paper, and so I would remove them.

9. Figure 5: I do not think the LC3 blots here are particularly helpful, because you don’t see a consistent phenotype following Torin treatment. In most panels shown, it looks like Torin has no obvious impact on LC3.

10. Figure 6: In the mutagenesis studies, if you combine the findings using ChII and the “QITSR”, it seems reasonable to conclude that the Q77 is less/not important, while the “ITSR” region is what dictates the resistance of CH077 Vpr to autophagy. That’s because the ChII is resistant to autophagy, and it contains only the “ITSR” region from CH077. In fact, when depicting that mutant in Figure 6D, the authors could shade the C-terminal region in yellow, just as they do when depicting the ChII construct.

11. Figure 6: I don’t like the designation “LAB-Vpr” because LAB is not an acronym—it’s just a short form of Lab-adapted. So, if the authors insist on referring to it as Lab, it should be “Lab-Vpr.” “TFV” is a proper acronym and so “TFV-Vpr” is logical. However, it would be more straightforward to just call the constructs NL43-Vpr and CH077-Vpr in the text and figures.

12. Figure 6: I would say that it is strange to show two replicates of the co-IP experiments in Figure 6F, but to show only one for Figure 6A-E. By indicating to the readers that NDP52 pulled down Lab-Vpr 50% of the time, they will be left asking “What percentage of the time does SQSTM1/TAX1BP1 pull down Lab-Vpr?”

13. Does Vpr co-immunoprecipitate with LC3? This is a logical question to ask since the authors observe colocalization between Vpr and LC3-GFP, and since they show co-IP between Vpr and certain autophagic receptors. Thus, Vpr may interact directly or indirectly with LC3.

PLOS authors have the option to publish the peer review history of their article (what does this mean? ). If published, this will include your full peer review and any attached files.

**Do you want your identity to be public for this peer review?** For information about this choice, including consent withdrawal, please see our Privacy Policy .

Reviewer #1: No

Reviewer #2: No

Reviewer #3: No

**Figure resubmission:**

**Reproducibility:**



---

## [Decision Letter · Decision Letter 1]

27 Jan 2026

PPATHOGENS-D-25-01899R1

HIV-1 Vpr is an autophagy target and its susceptibility to autophagy reduces virus transmission

PLOS Pathogens

Dear Dr. Serra-Moreno,

Thank you for submitting your manuscript to PLOS Pathogens. After careful consideration, we feel that it has merit but does not fully meet PLOS Pathogens's publication criteria as it currently stands. Therefore, we invite you to submit a revised version of the manuscript that addresses the points raised during the review process.

We look forward to receiving your revised manuscript.

Kind regards,

Edward M Campbell, PhD

Academic Editor

PLOS Pathogens

Susan Ross

Section Editor

PLOS Pathogens

Sumita Bhaduri-McIntosh

Editor-in-Chief

PLOS Pathogens

orcid.org/0000-0003-2946-9497

Michael Malim

Editor-in-Chief

PLOS Pathogens

orcid.org/0000-0002-7699-2064

**Additional Editor Comments:**

The reviews still have minor concerns regarding language that.can be resolved without any additional experimentation.

**Journal Requirements:**

**Reviewers' Comments:**

Reviewer's Responses to Questions

**Part I - Summary**

Reviewer #1: (No Response)

Reviewer #2: SUMMARY:

This revised manuscript presents interesting and potentially important findings on Vpr as an autophagy target. The authors have conducted extensive experimental work (10 main figures, 2 supplemental datasets) and demonstrate appropriate scientific rigor in acknowledging limitations. However, some key claims require closer alignment with the data, and clarifying the scope and limitations of certain conclusions would strengthen the manuscript. These issues can be addressed primarily through textual revisions to ensure claims precisely match the supporting data.

Reviewer #3: In this revision, the authors have performed extensive revisions to the text and figures (including of new experimental data) to address reviewers' points. The article has been improved as a result.

**Part II – Major Issues: Key Experiments Required for Acceptance**

Reviewer #1: The authors were very responsive and addressed all of my concerns. Either by new experiments or textual changes/discussion. Thank you for your efforts.

Reviewer #2: Most of the critiques can be addressed with textual Revision

1. Alignment of Transmission Claims with Assay Readout

The abstract and title emphasize effects on "HIV-1 transmission" (lines 28-31), and the GFP+/mCherry+ assay is framed as measuring "successful transmission."

The authors appropriately and honestly acknowledge (lines 786-793): "a defect in transmission for viruses harboring autophagy-sensitive Vpr proteins might be caused by a defect in infectivity...there are many steps in the replication cycle that can impact GFP and mCherry expression."

This caveat is scientifically sound but creates a discrepancy: the data show reduced GFP+/mCherry+ cells (a composite readout of multiple processes), whereas the claims specifically invoke "transmission" (particle transfer and entry).

Pls, revise the title, abstract, and key conclusions to more precisely reflect what was measured. Options include:

- Framing as "viral spread" or "infection/replication" rather than specifically "transmission."

- Explicitly stating throughout that the readout reflects composite effects

- Emphasizing that reduced GFP+/mCherry+ cells could result from multiple mechanisms

- Expand the discussion of this limitation beyond lines 786-793 to help readers understand the scope of the findings

- Acknowledge in the discussion that future work will be needed to dissect which specific step(s) are affected

2. Framing of Causality and Autophagy Dependence:

The manuscript uses pharmacological approaches (Torin2, VPS34IN, etc.) to implicate autophagy. The authors note difficulty with genetic rescue due to toxicity (lines 810-816).

Pharmacological data are valuable but have inherent limitations (off-target effects, pleiotropic responses). While ATG5KO cells are used for mechanistic studies, the transmission phenotype itself has not been demonstrated in autophagy-null genetic backgrounds.

Pls, acknowledge more explicitly that causality is established through pharmacological rather than genetic approaches for the transmission of the phenotype

- Discuss this limitation and note that while ATG5KO data support the mechanism, genetic validation of the transmission phenotype specifically would strengthen future work

- Note in the discussion that genetic approaches for the transmission assay will be important future directions

3. Ubiquitination Mechanism Discussion:

The ubiquitination data show complex patterns:

- TAK-243 rescues LAB-Vpr (Figure 10A)

- K27M is still degraded (Figure 10B)

- LAB-Vpr and TFV-Vpr show similar ubiquitination (Figure 10C)

- SQSTM1-ΔUBA still binds Vpr (Figure 10E-F)

The authors conclude that the effects are "ubiquitin-independent" (lines 751-769) and speculate about TAK-243's off-target effects (lines 735-749).

These findings are difficult to reconcile, and the current presentation may overstate what can be concluded.

The team should acknowledge more clearly that the ubiquitination results are complex and somewhat contradictory

- Present the TAK-243 rescue and K27M/ΔUBA data more evenhandedly rather than dismissing TAK-243 as off-target

- Revise the conclusion to state that ubiquitination's role remains unclear or requires further investigation

4. Primary Cell Context and Scope:

The mechanistic work primarily uses HEK293T cells, with limited validation in primary CD4+ T cells (Figures 1D-F, 4A-B).

HEK293T cells provide excellent experimental tractability but differ from physiologically relevant HIV-1 host cells.

The team should discuss the limitation that most mechanistic insights are from kidney epithelial cells

- Frame conclusions as "in our experimental system" rather than as universal HIV-1 biology

Reviewer #3: (No Response)

**Part III – Minor Issues: Editorial and Data Presentation Modifications**

Reviewer #1: (No Response)

Reviewer #2: 1. TFV Resistance Mechanisms:

The authors acknowledge (lines 672-678) that some TFV Vpr proteins lack the mapped resistance determinants, suggesting mechanistic diversity.

Pls clarify in the text that the CH077 mapping may represent one of potentially multiple resistance mechanisms

- Avoid overgeneralizing from one TFV to all TFVs

2. NDP52 Interaction Variability:

NDP52 co-IP showed 50% reproducibility (lines 425-427).

The team should acknowledge this variability more explicitly when discussing NDP52's role

MINOR ISSUES

- Sample sizes: State explicitly in figure legends the number of biological replicates for transmission experiments

- Statistical clarity: Clarify statistical approaches for transmission assays

- Consistency: Ensure consistent terminology (transmission vs. infection vs. viral spread) throughout

Reviewer #3: There remain a couple of issues that could be handled by reformatting and revising certain conclusions made by the authors:

1. Figure 4C appears incomplete since it does not contain the same experiment in WT cells (only ATG5 KO cells are shown). The MG132 and chloroquine treatments should also be performed with WT cells, since the authors need to show that Rapamycin/Torin treatment causes Vpr protein loss in WT cells, and that protein is recovered by chloroquine but not MG132. This would support the turnover of Vpr in lysosomes upon mTOR inhibition and would better complement the other data in the paper. Another option is to simply remove Figure 4C, because it does not add any appreciable value to the revised version of the paper.

2. The transmission data in the final figure still contains some of the least convincing functional data in the paper, despite it being revised. Even though the Vpr mutant that is resistant to autophagy confers virus with a slightly higher capacity for transmission, as the authors conclude, I think the effect sizes are rather small in both the 2D and 3D systems used. Also, as the authors admit, the use of the Nef mutant causes decreased virus transmission irrespective of which Vpr is expressed alongside it, making it difficult to use this set of conditions to discern how Vpr’s susceptibility to autophagy is related to virus transmission. It seems to me that the most appropriate conditions to compare to one another are VprLabNefMut and VprMutNefMut. That’s because by including the mutated Nef incapable of antagonizing autophagy (therefore the cellular autophagy response is intact), it should be easier to determine whether mutating the Vpr to an autophagy-resistant state impacts virus transmission. However, the authors do not perform a comparison and statistical analysis between those two conditions. That diminishes the utility of having NetMut in the figure. Nonetheless, by eye, the VprLabNefMut condition transmits as well as the VprMutNefMut in nearly all of the transmission systems studied (with the exception of the 3D total, where VprMutNefMut is relatively increased).Therefore, I think it is inappropriate to state “reduces virus transmission” in the title of the article. I would recommend that they revise the title to emphasize what their data best supports (that Vpr is targeted for degradation by autophagy).

PLOS authors have the option to publish the peer review history of their article (what does this mean? ). If published, this will include your full peer review and any attached files.

**Do you want your identity to be public for this peer review?** For information about this choice, including consent withdrawal, please see our Privacy Policy .

Reviewer #1: No

Reviewer #2: No

Reviewer #3: No

**Figure resubmission:**
---

## [Editor Report · Decision Letter 2]

20 Feb 2026

Dear Dr. Serra-Moreno,

We are pleased to inform you that your manuscript 'HIV-1 Vpr is targeted for degradation by autophagy' has been provisionally accepted for publication in PLOS Pathogens.

Best regards,

Edward M Campbell, PhD

Academic Editor

PLOS Pathogens

Susan Ross

Section Editor

PLOS Pathogens

Sumita Bhaduri-McIntosh

Editor-in-Chief

PLOS Pathogens

orcid.org/0000-0003-2946-9497

Michael Malim

Editor-in-Chief

PLOS Pathogens

orcid.org/0000-0002-7699-2064

Thank you for your continued proactive responsiveness to the reviewer comments. This is a strong study. Congratulations on its acceptance.
---

## [Editor Report · Acceptance letter]

Dear Dr. Serra-Moreno,

We are delighted to inform you that your manuscript, "HIV-1 Vpr is targeted for degradation by autophagy," has been formally accepted for publication in PLOS Pathogens.

Best regards,

Sumita Bhaduri-McIntosh

Editor-in-Chief

PLOS Pathogens

orcid.org/0000-0003-2946-9497

Michael Malim

Editor-in-Chief

PLOS Pathogens

orcid.org/0000-0002-7699-2064